# Kombucha and Water Kefir Grains Microbiomes’ Symbiotic Contribution to Postbiotics Enhancement

**DOI:** 10.3390/foods12132581

**Published:** 2023-07-02

**Authors:** Marina Pihurov, Bogdan Păcularu-Burada, Mihaela Cotârleț, Leontina Grigore-Gurgu, Daniela Borda, Nicoleta Stănciuc, Maciej Kluz, Gabriela Elena Bahrim

**Affiliations:** 1Faculty of Food Science and Engineering, Dunărea de Jos University of Galați, Domnească Street No. 111, 800201 Galați, Romania; marina.pihurov@ugal.ro (M.P.); bogdan.pacularu@ugal.ro (B.P.-B.); mihaela.cotarlet@ugal.ro (M.C.); leontina.gurgu@ugal.ro (L.G.-G.); daniela.borda@ugal.ro (D.B.); nicoleta.stanciuc@ugal.ro (N.S.); 2Department of Bioenergetics and Food Analysis and Microbiology, University of Rzeszow, 35601 Rzeszow, Poland

**Keywords:** water kefir grains, SCOBY membranes, co-fermentation, design of experiments, postbiotics

## Abstract

Wild artisanal cultures, such as a symbiotic culture of bacteria and yeasts (SCOBY) and water kefir grains (WKG), represent a complex microorganism consortia that is composed of yeasts and lactic and acetic acid bacteria, with large strains of diversity and abundance. The fermented products (FPs) obtained by the microbiome’s contribution can be included in functional products due to their meta-biotics (pre-, pro-, post-, and paraprobiotics) as a result of complex and synergistic associations as well as due to the metabolic functionality. In this study, consortia of both SCOBY and WKG were involved in the co-fermentation of a newly formulated substrate that was further analysed, aiming at increasing the postbiotic composition of the FPs. Plackett–Burman (PBD) and Response Surface Methodology (RSM) techniques were employed for the experimental designs to select and optimise several parameters that have an influence on the lyophilised starter cultures of SCOBY and WKG activity as a multiple inoculum. Tea concentration (1–3%), sugar concentration (5–10%), raisins concentration (3–6%), SCOBY lyophilised culture concentration (0.2–0.5%), WKG lyophilised culture concentration (0.2–0.5%), and fermentation time (5–7 days) were considered the independent variables for mathematical analysis and fermentation conditions’ optimisation. Antimicrobial activity against *Bacillus subtilis* MIUG B1, *Staphylococcus aureus* ATCC 25923, *Escherichia coli* ATCC 25922, and *Aspergillus niger* MIUG M5, antioxidant capacity (DPPH), pH and the total acidity (TA) were evaluated as responses. The rich postbiotic bioactive composition of the FP obtained in optimised biotechnological conditions highlighted the usefulness of the artisanal co-cultures, through their symbiotic metabolic interactions for the improvement of bioactive potential.

## 1. Introduction

Artisanal cultures are known as wild consortia of microorganisms which can grow on unconventional fermentation substrates, resulting in valuable products rich in bioactive compounds (biotics).

Briefly, SCOBY-based membranes are made of a natural consortium of microorganisms which work in mutualistic symbiosis due to a wide variety of species comprising mostly acetic acid bacteria (*Gluconacetobacter* ssp., *Acetobacter* ssp., and *Gluconobacter* ssp.) and yeasts (*Zygosaccharomyces* ssp., *Brettanomyces* ssp., and *Saccharomyces* ssp.), but also some lactic acid bacteria strains (*Lactobacillus* ssp.). The bacterial strains generate a polysaccharide stroma in which the yeasts are attached, causing the development of a membrane with a thickness of several centimetres at the liquid–air border [1,2]. This SCOBY-based membrane usually ferments black or green tea supplemented with sucrose. The fermentation took place over 7 to 10 days, at 20–25 °C, and as such a beverage with a rich postbiotic composition was achieved. It contained organic acids (acetic, lactic, malic, tartaric, citric, gluconic, and glucuronic), and other compounds (substances with antibiotic properties, ethanol, water-soluble vitamins, hydrolytic enzymes, and amino acids) [3,4].

WKG culture is typically composed of lactic acid bacteria (*Lactobacillus* ssp., *Streptococcus* ssp., and *Leuconostoc* ssp.), yeasts (*Saccharomyces* ssp. and *Dekkera* ssp.), and acetic acid bacteria, encapsulated in the polysaccharide dextran (and a limited concentration of levan) matrix, looking like hard granules, small, translucent, and irregular [5,6]. These grains are usually cultured in water and sucrose supplemented with dried fruits (often raisins or figs), afterwards producing a useful beverage [7]. After 2–3 days of fermentation at 20–25 °C, the FP contains a wide range of synthesised metabolites, including lactic acid, esters, glycerol, carbon dioxide, and acetic acid isoamyl acetate, as well as ethanol, ethyl octanoate, ethyl decanoate, and ethyl hexanoate [1,8].

The microbial strains’ symbiotic interaction in the artisanal consortium offers metabolite production in the FPs (pre-, pro-, post-, and para-probiotics). The diversity of the microbial community ensures the stability and safety of FPs against spoilage and pathogenic microorganisms [5]. The products fermented with artisanal cultures also exhibit valuable biological activities, including immunomodulatory, anti-inflammatory [9], antihypertensive, hepatoprotective, cholesterol-lowering, and antioxidant potential [10]. This study’s goal was the co-cultivation of SCOBY and WKG (lyophilised cultures) in a new formulated fermentation medium and the optimisation of the biotechnological parameters to produce FP with improved bioactive content.

## 2. Materials and Methods

### 2.1. Multiplication and Freeze-Drying of Water Kefir Grains as Starter Cultures

WKG (Medicer Bios, Bucharest, Romania) were grown on a specific medium based on sterile tap water supplemented with 10% (*w*/*v*) sugar and 1% (*w*/*v*) raisins, at 25 ± 1 °C, for a period of 48 h, under aerobic conditions. Further, the grains were firstly washed using Milli Q water, and then immersed in a fresh medium. The mixtures were incubated under the same conditions as mentioned before, with successive cultivation steps to allow the granules to multiplicate [11]. Furthermore, in order to assure the cryoprotection of the viable cells, the granules were washed with ultrapure water and supplemented with 10% (*w*/*v*) inulin and lyophilised at −80 °C (Christ Alpha 1-4 LD plus, Osterode am Harz, Germany). The cryodesiccated water kefir grains were ground and further stored at a temperature of 4 °C.

### 2.2. Multiplication of Kombucha’s Biofilm as a Starter Culture

The SCOBY-based membranes, purchased from a private household from the Republic of Moldova, were multiplied via successive cultivation in a sugar-based medium (7.5%, *w*/*v*) with the addition of 3% (*w*/*v*) infusion of black tea (Aaro Forstman Oy, Vantaa, Finland). The mixture was incubated for 10 days at room temperature (22 ± 2 °C). Then, 20% (*w*/*w*) sterile inulin powder was added to 100 g of biofilm that was previously divided in small pieces, the mixture being homogenised and freeze-dried [12]. The resulting biofilm was finally mortared and stored at 4 °C.

### 2.3. Experimental Design and Optimisation of WKG and SCOBY Co-Cultivation Process

#### 2.3.1. The Plackett–Burman Design to Select the Factors Influencing the Synergism between WKG and SCOBY

For the screening of the most significant parameters that influence the co-cultivation of WKG and SCOBY, FPs were prepared according to the Plackett–Burman experimental Design (PBD), by varying the following parameters: 1–3% (*w*/*v*) black tea leaves (the tea mixture being infused for 5 min in boiled water and cooled down until 90 ± 1 °C), 3–6% (*w*/*v*) raisins, 5–10% (*w*/*v*) sugar, pH = 6.30 (Table 1). After the medium’s sterilisation at 105 °C, for a period of 10 min, it was inoculated, after cooling, with the lyophilised starter cultures (0.2–0.5% *w*/*v*), and incubated under aerobic and stationary conditions for 5 to 7 days at 30 °C. After fermentation, the samples were analysed immediately.

The Minitab software took into consideration 3 central points and 6 factorial and thus generated 15 experimental variants. The responses that were regarded were the pH and titratable acidity, the antioxidant capacity, and the antimicrobial activities (against above-mentioned microorganisms).

#### 2.3.2. Optimisation of WKG and SCOBY Co-Cultivation via Response Surface Methodology (RSM)

After analysing the responses obtained from PBD experimental runs, the factors influencing the fermentation with WKG and SCOBY were identified, as follows: concentration of black tea and raisins, as well as the fermentation time. Subsequently, five variation levels were analysed for the independent variables (Table 2) included in the RSM analysis. For the statistical validation of the experimental models, a *p* value of <0.05 was regarded as being significant.

The factors that were not optimised (B, E, F) remained constant; respectively, a 5% sugar concentration (*w*/*v*), a 0.2% lyophilised SCOBY concentration (*w*/*v*), and a concentration of WKG of 0.2% (*w*/*v*).

### 2.4. The Evaluation of the Responses

#### 2.4.1. Acidifying Potential

The pH analysis was assessed with a digital pH meter (Mettler Toledo, FiveEasy F20, Greifensee, Switzerland).

The titratable acidity was expressed in Thörner degrees (°Th), using the AOAC method [13]. In brief, about 4 g of sample was weighed and distilled water was added to reach 50 mL, in volumetric flask. Aliquots of 10 mL were used for the NaOH 0.1N titration, using phenolphthalein, until the appearance of a weak pink colour.

#### 2.4.2. Evaluation of the Antifungal Properties of the FPs

The antifungal activity assessment was carried out on the indicator strain of *Aspergillus niger* MIUG M5, belonging to the MIUG Collection, from the Faculty of Food Science and Engineering, “Dunărea de Jos” University, Galati, Romania. The incubation of the mould strain was achieved at 25 °C, for 96 h, using the Yeast Glucose Chloramphenicol (YGC) Agar. The inoculum was obtained by the suspension of the spores in sterile saline solution (0.9% NaCl) at a concentration of 1 × 10^5^ spores/mL, by counting with the Thoma chamber. From the fresh FP, a volume of 1 mL was taken and dispersed in a Petri dish and thoroughly mixed with a volume of 20 mL of Potato Dextrose Agar (PDA) medium (Oxoid, England). After solidification, from the spore’s suspension a volume of 10 μL was inoculated in the centre of the plate and left for incubation at 25 °C for 4 days (96 h). The control was assessed under the above-mentioned conditions but without the FP addition. Subsequent to the incubation time, the diameters of the mould’s growth were determined and the inhibition ratio (*RI*) was determined using Equation (1) [14]:(1)$$RI=\frac{A_{c}-A_{t}}{A_{c}}\times 100$$
where *RI* represents the growth inhibition ratio, *A_c_* represents the mould growth’s diameter of the control sample, and *A_t_* the mould strain’s diameter on the FP-supplemented medium.

#### 2.4.3. Assessment of the FPs’ Antibacterial Activity

Antibacterial activity was tested against the indicator strains *Bacillus subtilis* MIUG B1, *Escherichia coli* ATCC 25922, and *Staphylococcus aureus* ATCC 25923, strains that were cultivated on Plate Count Agar (PCA) (Scharlau, Barcelona, Spain), respectively, and Mueller II Hinton agar (Biolab, Hungary) for 24 h, at 37 °C. Then, the colony was placed into the Nutrient Broth (for *B. subtilis*) or, respectively, Muller Hinton broth (for *E. coli* and *S. aureus*), with an overnight incubation at 37 °C. Afterwards, the bacterial inoculum was dimensioned spectrophotometrically (OD_600nm_) at 0.3, corresponding to a concentration around 2.4 × 10^8^ CFU/mL. Subsequently, a volume of 500 µL of bacterial suspension was added in the Petri dishes with the specific media for each bacterial strain in the wells made (8 mm diameter), and 100 μL of FP was added. Afterwards, the incubation of the plates took place at 37 °C, for 48 h, the inhibition zone being determined and expressed in mm [15,16].

#### 2.4.4. Evaluation of the Antioxidant Properties of the FPs

In order to extract the bioactive from the fermented medium, an ultrasound-assisted method (MRC. Ltd., Holon, Israel) was applied, considering an extraction time of 30 min at 40 °C, followed by centrifugation at 7000 rpm and 4 °C for 15 min. To obtain the 2,2-diphenyl-1-picrilhydrasyl (DPPH) radical scavenging potential (DPPH) solution, 4 mg of DPPH were transferred in 100 mL of HPLC-grade methanol (≥99.9%) and allowed to dissolve. The DPPH solution was prepared daily and stored in dark conditions [17,18]. From the supernatant, an aliquot of 0.1 mL sample was homogenised with 3.9 mL DPPH solution and kept in the dark for 90 min, and the absorbance was read at 515 nm [19,20].

Antioxidant activity was performed by adding the 2,2-diphenyl-1-picrilhydrasyl (DPPH) radical scavenging potential, allowing the antioxidant activity in μM TE/mL to be calculated on a calibration curve based on 6-hydroxy-2,5,7,8-tetramethylcroman-2-carboxylic acid (Trolox). The antioxidant activity was calculated according to the Formulas (2) and (3):RSA, % = [(A_m_ − A_p_)/A_m_] × 100(2)
μM TE/mL = (RSA, % − 3.3672)/0.3483(3)
where A_m_ was the absorbance of the control and A_p_ the absorbance of the sample analysed.

In brief, 4 mg of DPPH was transferred in 100 mL of HPLC-grade methanol (≥99.9%) and allowed to dissolve. The DPPH solution was prepared daily and stored in dark conditions [17,18].

### 2.5. Analysis of Organic Acids and Polyphenolic Content in the FP Obtained in Optimised Fermentation Conditions

#### 2.5.1. The Assessment of the Organic Acids

The determination was achieved using an HPLC system, Agilent 1200 (Agilent Technologies, Santa Clara, CA, USA), with a multi-wavelength detector (MWD) and a quaternary pump, autosampler, degasser, and a thermostat. The column was a Hamilton RPR X300 (250 × 4.1 mm, particle size 7 μm, Hamilton, Bonaduz, Switzerland), with a gradient elution composed of mobile phase A (KH_2_PO_4_, 20 mM, pH 2.4) and phase B—acetonitrile 90% (*v*/*v*) (ACN) [21]. The mobile phases’ mixtures had the following steps: min 0—80% A, min 10—40% A, min 12.5—40% A, min 12.6—80% A. The organic acids’ separation profile was achieved at 210 nm, injection volume of 20 μL, at 30 °C, with a 1.5 mL/min flow-rate [22]. The data acquisition was assessed with the ChemStation software.

The organic acids were identified and quantified based on external calibration curves using HPLC-grade organic acids’ standard solutions (Sigma Aldrich, Schnelldorf, Germany).

#### 2.5.2. Evaluation of Polyphenols

The Agilent 1200 high-performance liquid chromatography system (Agilent Technologies, Santa Clara, CA, USA) was applied to determine the polyphenolic composition of the analysed samples. Consequently, the compounds of interest were separated with a Synergi Max-RP-80Å column with a guard column (250 × 4.6 mm, particle size of 4 μm, Phenomenex, Torrance, CA, USA) using mobile phase A (ultrapure water: acetonitrile: formic acid = 87:3:10) and mobile phase B (ultrapure water: acetonitrile: formic acid = 40:50:10), with the following elution program: min 0—94% A, min 20—80% A, min 35—60% A, min 40—40% A, min 45—10% A. For the polyphenolic compounds’ separation, at 30 °C, a 20 μL volume was deployed into the column, at a 0.5 mL/min flow rate. The time of the method was 80 min and then the data were processed using ChemStation program version B.04.03 [14,23,24].

The polyphenols were identified and quantified simultaneously at the wavelengths of 280 nm and 320 nm, based on external calibration curves for the available polyphenolic HPLC-grade standards (Sigma Aldrich, Schnelldorf, Germany).

### 2.6. Statistical Analysis

The design of the experiments was assessed with the Minitab 17 software (v. 1.0, LLC, Pennsylvania State University, Centre County, PA, USA). One-way ANOVA and Tukey tests considering a 95% confidence interval (*p* < 0.05) were used to analyse the experimental results, which were considered as averages of triplicate measurements followed by standard deviation.

## 3. Results and Discussions

### 3.1. The Selection of the Most Important Parameters That Influenced the Fermentation Process with Artisanal Consortia via PBD Analysis

This strategy was applied for the fermentation process, aiming at designing the appropriate culture medium by adjusting the carbon (C) and nitrogen (N) sources, the C/N ratio, minerals, trace elements, growth factors, and the fermentation parameters. In the customised formulas for fermentation, the main source of C was considered, whereas the fresh or dried fruits provided the nitrogen [25]. The analysed parameters and the interactions between them could be evaluated objectively using statistical methods [26].

The statistical modelling with PBD generated 15 experimental combinations using the ranges of variation for the chosen factors, as follows: black tea concentration 1–3% (*w*/*v*), sugar concentration 5–10% (*w*/*v*), raisins concentration 3–6% (*w*/*v*), 5–7 days of fermentation, lyophilised SCOBY ranging from 0.2 to 0.5% (*w*/*v*), and WKG ranging from 0.2 to 0.5 % (*w*/*v*), respectively (Table 3).

Thus, the following results were obtained; 3.46–3.96 for pH and 20–225 °Th for titratable acidity, while for the antioxidant activity a value of 2.388–2.412 μM TE/mL was found, and 0.00–12.67 mm for the antibacterial activity inhibition zone against *E. coli*, 0.00–14.00 mm against *S. aureus*, 1.50–14.33 mm against *B. subtilis*, and an 82.06–100% inhibition zone for the antifungal activity.

The statistical models, based on some analysed responses, respectively, the antioxidant activity and the antibacterial activity against *S. aureus*, were validated in accordance with the regression coefficients higher than 80% and at a *p* < 0.05 value.

The main parameters that influenced the FPs’ antioxidant activity were the tea’s concentration (A), raisins’ concentration (C), and the time of fermentation (D), and their impact on the studied response variables is shown in the Pareto diagram (Figure 1a).

Analysing the results comparatively, it can be stated that the FPs that were involved in the co-culture were characterised by a higher bioactive capacity compared to the unfermented medium, due to the polyphenols present in the tea and the metabolites of yeasts and bacteria, including vitamins, organic acids, and extracellular enzymes that contribute to the structural and compositional changes during the fermentation of kombucha [27].

The product fermented with WKG had a high antioxidant potential on account of the presence of lactic and acetic acid bacteria, as well as yeasts, their metabolites, and cell lysis’ products that occurred during fermentation [28].

For the antibacterial activity against *S. aureus*, the significant factors were the concentration of tea (A) and the fermentation time (D), as the Pareto diagram in Figure 1a shows.

It is known that due to the production of post-biotics, FPs (including black tea substrate) have shown antibacterial activity of an 12–30.2 mm inhibition zone against several pathogens [29].

For the rest of the analysed responses, no validation of the model was achieved.

The ANOVA results from Table 4 showed the significant contributions for the concentration of tea (*p* = 0.004), the concentration of raisins, and for the fermentation time (*p* = 0.019). Furthermore, this statistical model can be validated based on in the non-significant lack of fit (*p* = 0.923) [30].

### 3.2. Optimisation of the Fermentation Process with Artisanal Co-Culture to Increase the FPs’ Functional Potential

The statistical results from the PBD allowed the selection of three important parameters: tea concentration, raisin concentration, and fermentation time. The other factors, namely, the concentration of SCOBY lyophilised culture (0.2%) and the concentration of WKG lyophilised culture (0.2%) at 30 °C for 5 days of fermentation, remained constant.

The amount of inoculum, the amount of sugar and fruit added, the medium composition, the amount of oxygen, and the time and temperature of fermentation were also mentioned as factors that determined the best fermentation of WKG and had an impact on the composition and properties of the FP [28].

Table 5 presents the experimental matrix obtained by the Central Composite Design (CCD) model that generated 20 experimental variants, with the corresponding analysed responses: pH, titratable acidity, antioxidant activity, and antibacterial and antifungal activities against the targeted strains.

Following the statistical analysis of the obtained results, two mathematical models, for pH and total acidity, were validated, with a probability value of 0.003, thus highlighting the factors with a significant interaction for each validated response (Table 6).

The interactions between the variables, as well as their impact on the response, can be visualised in the contour and surface graphs (Figure 2a–c), which highlight the correlation between the tea concentration, the fermentation time, the concentration of raisins, and the responses obtained for the validated models.

Analysing the above graphs, the acidification potential increased when increasing the concentration of raisins and decreasing the concentration of tea and the time of fermentation.

According to the CCD experimental data, the following values for the analysed responses were obtained: 3.42–3.97 for pH, 51.25–637.5 °Th, 2.006–2.395 μM TE/mL for the antioxidant potential, 0–6.67 mm antibacterial activity inhibition zone against *E. coli*, 0–7.67 mm against *S. aureus*, 5.17–18.5 mm against *B. subtilis*, and 70.12–100% inhibition for antifungal activity against *A. niger*.

Following the analysis of the validated models and the significant factors, an optimised FP with an increased bioactive activity was designed by the formulated medium based on 3.52% (*w*/*v*) raisins, 1.0% (*w*/*v*) black tea, and 5% (*w*/*v*) sugar, inoculated with 0.2% (*w*/*v*) WKG lyophilised culture and 0.2% (*w*/*v*) SCOBY lyophilised culture co-fermentation at 30 °C, for 5 days, under stationary aerobic conditions.

The validation models for pH and total acidity (titratable acidity) were then analysed. The experimental values ranged between the predicted values for a 95% confidence level. Also, the desirability of the model was 0.901, close to 1, which indicates that by following the chosen parameters favourable results for the analysed responses can be achieved (Table 7).

The FP obtained in optimised conditions was characterised by a high acidity of 375.83 °Th and a low pH of 3.30 compared to the control (unfermented sample), which had an acidity potential of 9.27 °Th and a pH value of 4.69. Also, the control showed no antibacterial activity for the antifungal inhibition calculated with a ratio of 2.68%. Therefore, the antioxidant activity was higher (2.507 μM TE/mL) due to the polyphenols from the tea. The analysis of the validated models showed that the optimisation of bio-processes improved the FPs’ acidification capacity.

### 3.3. Organic Acids and Polyphenols Content in the Optimised FP

#### 3.3.1. Organic Acids Content

Using the high-performance liquid chromatography technique, the compounds present in the unfermented medium (control) and the fermented one were quantified (Table 8).

Due to the symbiosis between yeasts and lactic acid bacteria, the development of the homopolysaccharide matrix from WKG and organic acid production were achieved. In this regard, yeasts helped bacteria by providing nitrogen as simple assimilable compounds (dipeptides, tripeptides and amino acids) through their proteolytic activity. Also, the carbon source had a key signification in the fermentative capacity of the WKG [25]. The association and competition between the bacteria and yeasts in kombucha were unique, leading to chained reactions from various metabolites, including up to 6.4 g/L in acetic and lactic acids, and up to 0.5 g/L in citric, gluconic, malic, and succinic acids [31]. The consortium members’ cooperative association is well established. Less than 30% of the consortium is made of lactic acid bacteria strains, which are recognised for producing both lactic and also gluconic acids, which contribute to the antibacterial and antioxidant characteristics of the FP [32].

Among the identified short-chain fatty acids, acetic acid is characteristic for SCOBY fermentation, also being produced in small amounts as a postbiotic of the WKG consortium. As such, the acetic acid concentration increased from 4.34 mg/mL to 8.72 mg/mL. Butyric acid is not frequently found in kombucha-based drinks, but still its presence may occur, as Uţoiu et al., 2018 reported; after 5 days of fermentation, 0.14 g/L butyric acid was determined [33], compared to the present study, where the amount of butyric acid increased from 37.90 mg/mL to 45.81 mg/mL. Isovaleric acid is a volatile compound that contributes to the flavour of the FP, being the result of the interaction between acetic acid bacteria and yeasts (e.g., *Acetobacter indonesiensis* with *Brettanomyces bruxellensis*). The literature highlighted a concentration of up to 0.007 mg/mL; instead, the present study reported an amount of 0.88 mg/mL in the FP.

Previously, in an FP obtained by co-fermentation with milk kefir grains and SCOBY, some organic acids such as lactic acid, acetic acid, citric acid, isovaleric acid, and butyric acid, which presented the following concentrations, respectively, of 24.39 mg/mL, 25.21 mg/mL, 5.77 mg/mL, 4.36 mg/mL, and 67.33 mg/mL, were synthesised by the artisanal cultures [12]. Therefore, the lactic and citric acids were not identified in the optimised fermented product’s composition obtained by fermentation of the formulated medium with a multiple starter culture, based on WKG and SCOBY microbiota; the result can be attributed to the synergistic functionality of the consortia in tested conditions, in correlation with the chemical composition of the fermentation substrate.

#### 3.3.2. Content of Polyphenols and Flavonoids

The major bioactive compounds identified in the product obtained in optimised conditions were caffeic acid, 255.64 μg/mL; rutin trihydrate, 568.93 μg/mL; and epicatechin, 1135.69 μg/mL, whereas the caffeic acid was found in a lower concentration in the control, respectively, 16.80 μg/mL, this bioactive compound being specific to black tea (Table 9). Gallic acid and isorhamnetin, ferulic, and chlorogenic acids were present in smaller concentrations. Some compounds have been identified by Vázquez-Cabral et al., 2017, in a kombucha beverage, e.g., myricetin, 0.184 mg/L; gallic acid, 54.396 mg/L; caffeic acid, 16.213 mg/L; chlorogenic acid, 0.539 mg/L; epicatechin, 142.62 mg/L; and rutin, 4.245 mg/L. Our experimental data were confirmed by other results from similar works, that reported, in a fermented product with WKG, compounds such as chlorogenic acid, caffeic acid, tannins, vitamins C and D, glucosides, and various enzymes including lipase, amylase, and protease [34].

Previously, in our research regarding co-fermentation with SCOBY and milk kefir grains, in the sample obtained under optimised conditions, several compounds were quantified: gallic acid ≅ 71 μg/mL, epicatechin ≅ 1063 μg/mL, caffeic acid ≅ 315 μg/mL, quercetin ≅ 18 μg/mL, apigenin ≅ 0.22 μg/mL, and isorhamnetin ≅ 3 μg/mL [12].

Following the statistical and mathematical modelling analysis, an FP with improved bioactive properties was obtained. In the tested biotechnological conditions, the main independent variables with an influence on the quality of the FP turned out to be the concentration of tea, the fermentation time, and the concentration of raisins. Thus, the optimised fermentation conditions were: (i) composition of the medium: 3.52% (*w*/*v*) raisins, 1.0% (*w*/*v*) black tea, 5% (*w*/*v*) sugar in sterilised tap water; (ii) inoculum: 0.2% (*w*/*v*) lyophilised culture of WKG and 0.2% (*w*/*v*) lyophilised culture of SCOBY; and (iii) fermentation process: aerobic conditions, in a stationary system, at a temperature of 30 °C, for 5 days.

According to these biotechnological conditions, the obtained FP presented a high acidity potential of 375.83 °Th, and a 3.25 pH value. The organic acids were also highlighted in different concentrations; acetic—8.72 mg/mL, butyric—45.81 mg/mL, isovaleric—0.88 mg/mL, respectively; polyphenolic compounds such as phenolic acids: caffeic—255.64 μg/mL, gallic—39.68 μg/mL, ferulic—0.36 μg/mL, and chlorogenic—0.25 μg/mL; and flavonoids derived from quercetin: rutin trihydrate—568.93 μg/mL, isorhamnetin—11.94 μg/mL, and epicatechin—1135.69 μg/mL. The presence of these compounds demonstrates the functional potential of the FP.

## 4. Conclusions

The obtained results confirmed the possibility of using multiple SCOBY and WKG starter cultures to ferment a newly formulated black tea, raisin, and sugar-based medium to improve the postbiotic composition of the FP. The variant of FP with the increased bioactive potential was obtained following statistical techniques for the selection of the parameters (independent variables) and the optimisation of the process. The analysed responses demonstrated the ability of the bacteria and yeasts from the SCOBY and WKG microbiome to work in symbiosis. The preservation and usage of the artisanal cultures as freeze-dried cultures ensured the stability of the strain’s functionality. This study demonstrated the versatile metabolism and synergism of the wild microorganisms (bacteria and yeasts) from the multiple consortia. The idea of using these artisanal cultures for the co-fermentation of the unconventional substrates demonstrated their employment for multiple applications. Thus, by variation of the fermentation parameters and exploitation of PBD and CCD tools it is possible to obtain FPs with different compositions and bioactive properties to be used as ingredients for food and feed formulation.

## Figures and Tables

**Figure 1 foods-12-02581-f001:**
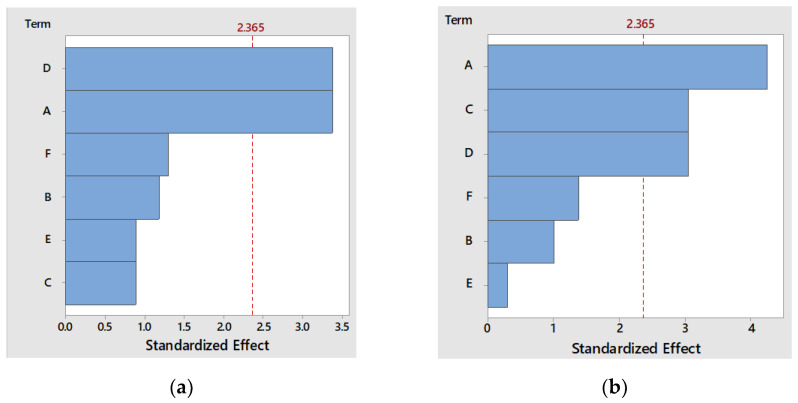
Pareto diagram of the independent variables’ effect studied on (**a**) antibacterial capacity against *S. aureus* strain and (**b**) antioxidant activity of FPs.

**Figure 2 foods-12-02581-f002:**
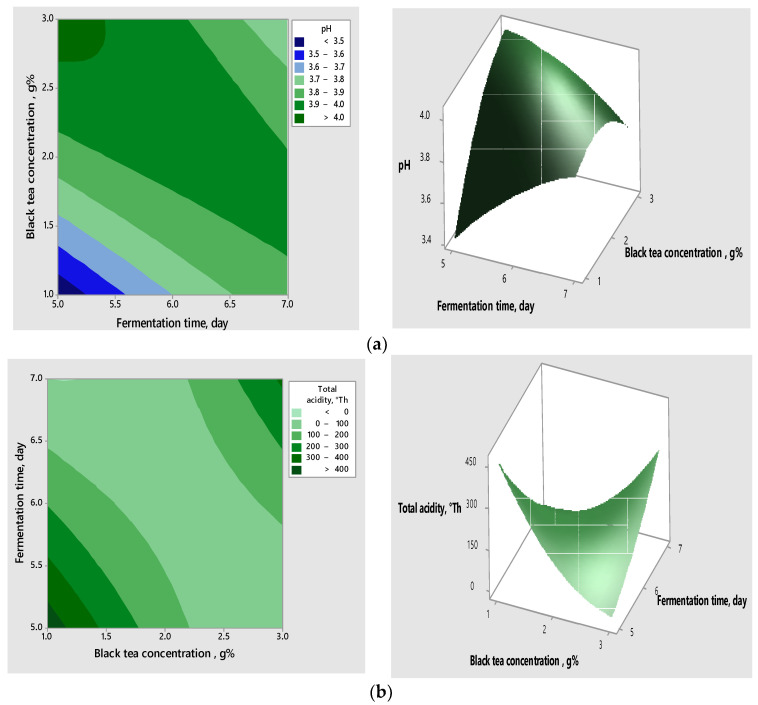
Contour graphs (**left**) and surface graphs (**right**) for correlative effect of the significant variables on pH (**a**) and total acidity (**b**,**c**).

**Table 1 foods-12-02581-t001:** Independent variables and the PBD variation ranges.

Independent Variables	Minimum Value	Maximum Value
	−1	+1
A, black tea concentration, % (*w*/*v*)	1.0	3.0
B, sugar concentration % (*w*/*v*)	5.0	10.0
C, raisins concentration % (*w*/*v*)	3.0	6.0
D, time of fermentation, days	5.0	7.0
E, freeze-dried SCOBY concentration, % (*w*/*v*)	0.2	0.5
F, freeze-dried concentration WKG, % (*w*/*v*)	0.2	0.5

**Table 2 foods-12-02581-t002:** Variation levels of the independent variables in RSM.

Independent Variables	Variation Levels				
	−1	0	+1	−α	+α
A, black tea concentration, % (*w*/*v*)	1.00	2.00	3.00	1.39	2.61
C, raisins concentration % (*w*/*v*)	3.00	4.50	6.00	3.58	5.42
D, fermentation time, days	5.00	6.00	7.00	5.39	6.61

**Table 3 foods-12-02581-t003:** The PBD of experiments and the corresponding responses obtained based on the independent variables’ variation.

Run	Independent Variables						Responses						
	A *	B	C	D	E	F	pH	Total Acidity, °Th	Antioxidant Activity, µM TE/mL	Antibacterial Activity against *E. coli*, mm	Antibacterial Activity against *S. aureus*, mm	Antibacterial Activity against *B. subtilis*, mm	Antifungal Activity against *A. niger*, RI %
1	1.0	10.0	6.0	7.0	0.2	0.5	3.85	37.50	2.411	0.00	0.00	5.00	100.00
2	3.0	5.0	6.0	7.0	0.2	0.5	3.94	40.00	2.404	0.00	0.00	7.00	82.96
3	2.0	7.5	4.5	6.0	0.35	0.35	3.85	51.25	2.412	0.00	0.00	5.83	100.00
4	1.0	5.0	6.0	7.0	0.5	0.2	3.87	50.00	2.409	0.00	0.00	3.83	100.00
5	2.0	7.5	4.5	6.0	0.35	0.35	3.83	43.75	2.412	0.00	0.00	5.33	100.00
6	1.0	10.0	6.0	5.0	0.5	0.2	3.85	41.25	2.398	0.00	14.00	4.17	88.34
7	1.0	5.0	3.0	7.0	0.5	0.5	3.82	45.00	2.401	0.00	0.00	1.50	100.00
8	3.0	10.0	6.0	5.0	0.5	0.5	3.96	38.75	2.393	0.00	0.00	7.33	100.00
9	2.0	7.5	4.5	6.0	0.35	0.35	3.88	36.25	2.401	0.00	0.00	5.50	100.00
10	1.0	10.0	3.0	5.0	0.2	0.5	3.87	20.00	2.394	0.00	11.67	2.50	100.00
11	1.0	5.0	3.0	5.0	0.2	0.2	3.46	225.00	2.398	12.67	12.33	14.33	82.06
12	3.0	5.0	3.0	5.0	0.5	0.5	3.96	56.25	2.388	0.00	0.00	6.00	100.00
13	3.0	10.0	3.0	7.0	0.2	0.2	3.91	62.50	2.388	0.00	0.00	7.17	100.00
14	3.0	10.0	3.0	7.0	0.5	0.2	3.88	56.25	2.390	0.00	0.00	6.33	83.86
15	3.0	5.0	6.0	5.0	0.2	0.2	3.92	58.75	2.388	0.00	0.00	6.00	100.00

* A—tea concentration, % (*w*/*v*); B—sugar concentration, % (*w*/*v*); C—raisins concentration % (*w*/*v*); D—time of fermentation, days; E—SCOBY culture’s concentration % (*w*/*v*); F—WKG culture’s concentration, % (*w*/*v*).

**Table 4 foods-12-02581-t004:** Antioxidant activity based on the ANOVA test.

Source	DF	Adj SS	Adj MS	F Value	*p* Value
Pattern	7	0.001001	0.000143	8.38	0.006
Linear	6	0.000677	0.000113	6.61	0.013
Concentration of tea, g%	1	0.000308	0.000308	18.03	0.004
Sugar concentration, g%	1	0.000018	0.000018	1.03	0.343
Concentration of raisins, g%	1	0.000159	0.000159	9.30	0.019
Time of fermentation, days	1	0.000159	0.000159	9.30	0.019
Concentration of kombucha, g%	1	0.000002	0.000002	0.09	0.774
Concentration of water kefir granules, g%	1	0.000032	0.000032	1.89	0.211
Curvature	1	0.000324	0.000324	19.01	0.003
Error	7	0.000119	0.000017		
Lack-of-Fit	5	0.000043	0.000009	0.22	0.923
Pure error	2	0.000077	0.000038		
Total	14	0.001120			

**Table 5 foods-12-02581-t005:** CCD with the analysed responses correlating with the independent variables’ variation in the RSM analysis.

Independent Variables				Responses						
Run	A *	C	F	pH	Total Acidity, °Th	Antioxidant Activity, µM TE/mL	Antibacterial Activity against *E. coli*, mm	Antibacterial Activity against *S. aureus*, mm	Antibacterial Activity against *B. subtilis,* mm	Antifungal Activity against *A. niger*, RI %
1	2.00	4.50	6.00	3.92	62.50	2.384	0.00	0.00	7.33	100.00
2	1.39	5.42	5.39	3.42	637.50	2.385	6.67	5.50	18.50	70.12
3	2.00	4.50	6.00	3.90	75.00	2.395	0.00	0.00	7.17	100.00
4	2.61	5.42	5.39	3.94	158.75	2.381	0.00	0.00	7.00	100.00
5	2.61	3.58	5.39	3.91	100.00	2.391	0.00	0.00	8.50	89.24
6	1.39	3.58	5.39	3.67	137.50	2.381	5.67	7.67	10.33	100.00
7	1.39	3.58	6.61	3.83	87.50	2.388	0.00	0.00	7.83	100.00
8	2.00	4.50	6.00	3.92	63.75	2.379	0.00	0.00	6.50	100.00
9	1.39	5.42	6.61	3.82	112.50	2.381	0.00	0.00	5.17	100.00
10	2.61	5.42	6.61	3.91	143.75	2.389	0.00	0.00	6.67	100.00
11	2.00	4.50	6.00	3.89	113.75	2.386	0.00	0.00	7.33	85.66
12	2.61	3.58	6.61	3.91	140.00	2.372	0.00	0.00	7.83	75.30
13	2.00	4.50	7.00	3.90	57.50	2.372	0.00	0.00	7.17	89.24
14	2.00	3.00	6.00	3.95	51.25	2.354	0.00	0.00	6.17	100.00
15	2.00	4.50	5.00	3.95	56.25	2.373	0.00	0.00	7.00	100.00
16	2.00	4.50	6.00	3.97	53.75	2.357	0.00	0.00	7.17	89.64
17	1.00	4.50	6.00	3.80	138.75	2.372	0.00	0.00	5.33	88.84
18	2.00	6.00	6.00	3.92	140.00	2.006	0.00	0.00	8.00	100.00
19	2.00	4.50	6.00	3.92	112.50	2.375	0.00	0.00	8.50	100.00
20	3.00	4.50	6.00	3.92	87.50	2.372	0.00	0.00	7.83	100.00

* A—black tea’s concentration, % (*w*/*v*); C—raisins’ concentration, % (*w*/*v*); F—fermentation time, days.

**Table 6 foods-12-02581-t006:** Validation of the interaction of significant independent variables.

Response	Variables with Significant Interaction	*p* Value
pH	1. Black tea’s concentration, g%, and Fermentation time, days	0.003
Total acidity, °Th	1. Black tea’s concentration, g%, and Raisins’ concentration, g%	0.013
	2. Black tea’s concentration, g%, and Time of fermentation, days	0.003
	3. Raisins’ concentration, g%, and Time of fermentation, days	0.006

**Table 7 foods-12-02581-t007:** Validation of the models.

Response	95% Confidence Level Range	Experimental Value
pH	3.30–3.70	3.30 ± 0.01
Titratable acidity	55.5–444.6	375.83 ± 0.02
Desirability of the model	0.901	

**Table 8 foods-12-02581-t008:** Content of organic acids in FP vs. unfermented medium.

Organic Acid	Concentration	
	Unfermented Medium, mg/mL	Fermented Product, mg/mL
Acetic acid	4.34 ± 0.01 ^B^	8.72 ± 0.12 ^A^
Butyric acid	37.90 ± 0.42 ^B^	45.81 ± 0.17 ^A^
Isovaleric acid	ND *	0.88 ± 0.12 ^A^

^A,B^—significant differences in the obtained values compared to the control (*p* < 0.05). * ND—not determined.

**Table 9 foods-12-02581-t009:** Content of bioactive compounds in the FP vs. unfermented medium.

Bioactive Compound	Unfermented Medium, μg/mL	Fermented Product, μg/mL
Gallic acid	ND	39.68 ± 2.56 ^a^
Caffeic acid	16.80 ± 0.02 ^b^	255.64 ± 54.77 ^a^
Chlorogenic acid	2 ± 0.00 ^a^	0.25 ± 0.02 ^b^
Ferulic acid	ND	0.36 ± 0.00 ^a^
Rutin trihydrate	ND	568.92 ± 50.06 ^a^
Epicatechin	ND	1135.59 ± 44.76 ^a^
Isorhamnetin	ND	11.04 ± 0.96 ^a^

The results are expressed as mean ± standard deviation. Values with different superscript letters in the same row indicate significant differences between samples (*p* < 0.05). ^a, b^—significant differences between the control and fermented sample.

## Data Availability

Data are contained within the article.

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
