# Peer review of "Kombucha and Water Kefir Grains Microbiomes’ Symbiotic Contribution to Postbiotics Enhancement"

_foods, 2023, doi:10.3390/foods12132581_

Round 1
Reviewer 1 Report
This study aims to identify postbiotics, antioxidant and anti microbial activities of a Symbiotic Culture of Bacteria and water kefir grains. The general idea and the experimental design are good. However, there are lack of information on the expected metabolites produced by this combination. Moreover, the microbiome of this symbiotic community was not identified. This may restrict the repeatability of the data, as the individual microbial strain of this symbiotic is not fully known.
Other comments:
Abstract
Line 10: SCOBY-based membranes (Symbiotic Culture of Bacteria and Yeasts), please note that the abbreviation which should be between brackets not vise versa.
Please, mention the best recommended combination obtained in the study
Line 82: add reference for the used procedure.
Table 8: Why lactic acid is not one of the detected organic acids, however the symbiotic includes lactic acid bacteria, which is supposed to produce considerable amounts of this organic acid. Moreover, give an explnation for the lack of detection of some common organic acids such as formic, citric,….which were detected in provious studies but not in your study (Lines 363-366)
Table 9: It is not clear to me why some phenolic appears at specific wavelength, while disappeared at other wavelength?? This does not provide informative data, for example, what is the meaning of detecting Caffeic acid at 320nm in unfermented product and its disappearance at the same wavelength in the fermented product. This table needs to be restructured using ideal wavelength and ideal hplc detector for each phenolic compound.
Line 395: for the optimized conditions: Mentioning the percentage of the WGC or SCOPY is not enough. The number of the live bacteria/yeast of the mentioned amount should be known. 0.2% of these cultures should not always contain the same numbers of microbial community and/or microbial strains.
Minor revision for the quality of English is needed
Author Response
Cover letter
June 28th, 2023
Manuscript ID: foods-2474376_R1
Type: Article
Title: Kombucha and water kefir grains microbiomes' symbiotic contribution
upon postbiotics enhancement
Authors: Marina Pihurov, Bogdan Păcularu-Burada, Mihaela Cotârleț,
Leontina Grigore-Gurgu, Daniela Borda, Nicoleta Stănciuc, Maciej Kluz*,
Gabriela Bahrim*
Dear Editors,
Dear Reviewers,
The authors kindly appreciate your efforts in reviewing and improving this manuscript.
This study's goal was the co-cultivation of multiple consortia of microorganisms of SCOBY-membrane and water kefir grains (freeze-dried cultures) in a new formulated fermentation medium and optimization of the biotechnological parameters in order to produce a fermented product with improved bioactive content.
Taking into consideration the valuable suggestions of reviewers the manuscript was revised, and we consider that it was significantly improved.
The modifications were made with Track Changes in the manuscript and highlighted as follows:
Reviewer 1 – blue font color
Reviewer 2 - green font color
Reviewer 3 -
The manuscript was carefuly revised regarding the plagiarism and for improving English language. All the modification were marked in red and with Track Changes.
Please see below the responses to the reviewers’ comments.
Kind regards,
The authors

Reviewer 2 Report
Comments to author
This study effectively demonstrated the process of optimizing the co-culture medium of WKG and SCOBY using PBD and CCD. And the optimized culture medium improved the antibacterial and antioxidant activity of the FPs. But there are a few questions that need to be considered.
Introduction
What are the scientific assumptions of this study? And why choose SCOBY-based membranes and WKG culture for co-fermentation? Besides both being able to produce postbiotics, are there any other commonalities that can be co-cultured together.
L70 Were water kefir grains cultivated under aerobic or anaerobic conditions?
L117 Is the sample a solid-liquid mixture?
L118 What is a necessary volume?
L212 Delete ‘be’.
L295-298, 306-307 There is no information on raisin concentration in Figure 2.
Conclusion
The experiment demonstrated the biological activity of the FPs well and optimized the culture medium, but the optimized culture medium was only suitable for co-cultivation of WKG and SCOBY. However, these two types of inoculants are not stable products, so the culture medium optimized by the author cannot be applied to inoculants from other regions or products. What is the recommendation or implementation in the next study?
Author Response

(The authors gave the same response as above.)

Round 2
Reviewer 1 Report
No further comments
The quality of English is acceptable